# Nature-Based Interventions Targeting Elderly People’s Health and Well-Being: An Evidence Map

**DOI:** 10.3390/ijerph21010112

**Published:** 2024-01-19

**Authors:** Giulia Catissi, Gabriela Gouveia, Roberta Maria Savieto, Cristiane Pavanello Rodrigues Silva, Raquel Simões de Almeida, Gustavo Benvenutti Borba, Kaue Alves Rosario, Eliseth Ribeiro Leão

**Affiliations:** 1Albert Einstein Israeli Faculty of Health Sciences, Hospital Israelita Albert Einstein, São Paulo 05651-901, Brazil; giulia.catissidelima@einstein.br; 2A Beneficência Portuguesa de São Paulo, São Paulo 01323-001, Brazil; ga_gouveia@hotmail.com; 3Hospital Israelita Albert Einstein, Education and Research Center, São Paulo 05651-901, Brazil; roberta.savieto@einstein.br; 4Center for Health Technology and Services Research (CINTESIS), Santa Maria Health School, 4000-088 Porto, Portugal; cristiane.silva@santamariasaude.pt; 5Psychosocial Rehabilitation Laboratory (LabRP-CIR, ESS), Polytechnic of Porto, 4200-072 Porto, Portugal; afa@ess.ipp.pt; 6Department of Electronics-DAELN, Graduate School on Biomedical Engineering—PPGEB, Federal University of Technology-Paraná—UTFPR, Curitiba 80230-901, Brazil; gustavobborba@utfpr.edu.br (G.B.B.); kauealves@alunos.utfpr.edu.br (K.A.R.)

**Keywords:** healthy aging, health promotion, nature

## Abstract

Background: Healthy aging encompasses more than the absence of disease, emphasizing the preservation of functional abilities for enhanced well-being and quality of life. Nature-based interventions are scientifically proven contributors to healthy aging. Objective: To develop an evidence map showcasing nature-based interventions targeting older individuals’ health and well-being. Methods: The evidence map was developed through critical analysis of systematic reviews and clinical trials utilizing the tools AMSTAR2 and CONSORT. A systematic search spanning the past decade was conducted across databases: Cochrane, SCOPUS, PubMed, Web of Science, Embase, and LILACS. Results: Twelve articles met the eligibility criteria. Nature-based interventions such as forest bathing, hiking, therapeutic gardens, virtual reality, and forest sounds were identified. Outcomes were categorized into physical aspects (cardiovascular and pulmonary; neuro-immuno-endocrinological) and mental/behavioral aspects. The final map integrated interventions, outcomes, and quality assessments. Conclusions: The survey highlights the positive impact of nature-based interventions on the health of the elderly. This study provides insights across various domains, fostering the development of programs and policies in management to promote healthy aging. Regarding healthcare, it encourages discourse among professionals regarding the integration of nature-based practices for equitable care in both individual and group settings. Furthermore, it underscores the need for research in the Southern Hemisphere, particularly in Brazil, where the study was conducted.

## 1. Introduction

Population aging is a recent global phenomenon, which has only been achieved due to advances in medical sciences and healthcare over the past two centuries, as well as the general improvement in living and working conditions, with significant demographic, biological, social, economic, and behavioral transformations [1]. With unprecedented numbers in human history, there is a transition in the demographic structure of the European Union, which in 2020 had 20.6% of the population being over 65 years old, with a projected increase to 29.5% in 2050 [2]. Specifically in Portugal, also in 2020, the number of older people corresponded to 22.4% of the country’s population [3]. Between 2011 and 2019, there was a reduction in the rate of live births, accompanied by an increase in the fertility rate. However, this increase was insufficient to ensure the replacement of generations. According to demographic projections, Portugal is expected to experience a population decline by 2080, decreasing from 10.3 million residents in 2019 to 8.2 million. At the national level, projections indicate a decrease in the proportion of young people in the total population and an increase in the proportion of individuals aged 65 or over. This shift is projected to result in an almost doubling of the aging rate [4]. In Brazil, although it is known as a “young country”, there is an inversion of the age pyramid, characterized by a significant increase in the elderly population. There has been a significant increase in life expectancy, which was 76.3 years in 2018 and is expected to reach 81.3 years in 2050 [5]. This increase directly impacts the absolute number of older adults in the Brazilian population.

In 2010, the number of people over 60 years of age was 14.2 million, with projections of 41.5 million in 2030 and 73.5 million in 2060, so that in 2025, Brazil should be the sixth ranking country in the world in terms of elderly population [5]. Such changes in the age structure of the population result in a continued and strong demographic aging, emphasizing the need to create and strengthen public policies that are aimed at promoting active and healthy aging. This involves not only the absence of diseases but also the preservation of functional capacity and the maintenance of well-being and quality of life [6].

According to the World Health Organization (WHO), elements such as access to economic resources, levels of education, physical and social environment, working conditions, access to healthcare services, and individual behaviors are considered key determinants of health. They constitute a complex set of factors that directly influence the health status of populations. Understanding these determinants is crucial for developing effective public policies aimed at improving the health of populations, especially the elderly, whose aging process involves multifactorial elements. Therefore, integrating these determinants into public health programs and policies addresses immediate health needs and establishes structures to address the root causes of diseases. The aim is comprehensive health promotion, targeting not just disease absence but physical, mental, and social well-being [7].

In Brazil and globally, there are legislations and recommendations that are aimed at strengthening such public policies. For instance, Brazilian Federal Law No. 10741 of 1 October 2003 asserts that it is the duty of the State and society to preserve the physical and mental health of older adults [8]. Additionally, the World Health Organization’s document, “Global Strategy and Action Plan on Aging and Health” outlines global strategies to support healthy aging [9]. As part of global initiatives, the known factors contributing to healthy aging and maintaining a quality of life include regular physical exercise, a balanced diet, pleasant social interactions between family and friends, regular sleep, full exercise of autonomy, periodic check-ups, activities that promote a sense of well-being, and stress reduction [10,11].

In this context, nature-based interventions are already pointed out as sources of several health benefits for the general population, with particular relevance for the well-being of the elderly. These benefits include the improvement of mental health, with a reduction in stress, anxiety, and depression [12]; positive effects on mood [13]; the reduction in blood pressure levels in hypertensive patients [14]; the improvement of interpersonal relationships [15]; the restoration of attention and memory [16]; the improvement of the immune system [17]; and the management of chronic pain [18]. Additionally, the known influence of contact with nature on mental health, coupled with its economic impact on public coffers, is noteworthy. In Australia, the mental health benefits derived from regular visits to natural areas alone generate the relevant savings of USD 6 trillion annually. On the other hand, if the population does not have the opportunity to visit parks and natural areas, the increase in mental health treatment costs can rise from 10% to 17.5% of the Gross Domestic Product [19].

Therefore, sustaining activities that foster a consistent human–nature relationship is not only directly linked to health promotion but also has indirect economic implications. This reinforces the necessity of expanding knowledge and actions on this topic in order to influence macro-level management that recognizes the environments that are essential for the well-being of the elderly and promotes healthy aging.

Another niche that can benefit from deepening our knowledge on the impact of nature-based interventions on the health of older people is tourism. In Europe, viewing tourism in natural areas as a tool for active aging presents an opportunity for the social and economic development of large areas, where the preservation of the environment is also a source of health and well-being [20].

However, sectors such as health, economics, and social well-being, which stand to benefit from the incorporation of nature as a tool for promoting healthy aging, currently lack sufficient scientific evidence on the topic. Existing literature on Nature and Health in the context of aging focuses on pre-existing conditions and the benefits of nature contact as an indirect form of treatment, often associated with pharmacological, procedural, or interventional treatment [21,22,23].

Nature-based interventions refer to numerous care modalities, such as forest bathing, forest therapy, therapeutic gardens, and other activities that are developed in natural settings. This diversity poses challenges for decision making, both for health professionals tasked with selecting the most effective interventions for older adults and stakeholders making investment decisions related to natural areas that are targeted at this public (in terms of infrastructure and tourism). Additionally, formulating appropriate public policies that effectively integrate natural environments with healthcare presents a complex challenge.

Despite the recognition that individuals, irrespective of their surroundings (urban, peri-urban, rural), gain greater health and well-being benefits from contact with biodiverse and perceived natural areas [24], we still face a lack of comprehensive elements to systematize actions targeting the elderly across diverse contexts.

Thus, considering the growing demand for advancing the implementation of nature-based interventions promoting the health and well-being of the elderly, we present an evidence map facilitating access to information and scientific evidence in health within this domain. In both clinical and management contexts related to natural spaces and services, making well-informed decisions necessitates access to robust, high-quality evidence. However, not all evidence is equally convincing or reliable, and a comprehensive synthesis on the topic is currently lacking for the older population. Our evidence map aims to address this gap by identifying, describing, and organizing available evidence on nature-based interventions targeting older people. It serves as a valuable resource for informed decision making, offering researchers, managers, professionals, and students in the fields of health, natural environments, and related areas access to the best knowledge and evidence available. It should also be emphasized that an evidence map can facilitate the translation of scientific knowledge and its incorporation into the real world and point out gaps for the development of future research [25,26].

## 2. Materials and Methods

### 2.1. Study Design

The study is an evidence map, which consists of a systematic search of a broad field to identify what is known about a particular topic (nature and characteristics), revealing gaps in knowledge and/or future research needs, which is particularly important for interventions that may be implemented without sufficient evidence. It is a way of presenting results in a user-friendly format, usually as a visual figure or graph or a searchable database [25].

To build the map, the authors followed the following steps: (1) definition of the scope to develop a structure that represents the range of interventions and important results of the available outcomes; (2) establishment of the search strategy for each database used; (3) exclusions of articles that did not fit the inclusion criteria of the study; (4) quality assessment of the included articles using AMSTAR2 [27] for systematic reviews and a tool based on CONSORT [28] for clinical trials; and (5) elaboration of the evidence map. The authors also followed the recommendations of the Preferred Reporting Items for Systematic Reviews and Meta-Analysis Extension for Scoping Reviews (PRISMA) [29].

### 2.2. Eligibility Criteria

The authors included systematic review articles and randomized controlled trials that focused on nature-based interventions in a variety of modalities focused on the health and/or well-being of older adults. The authors defined systematic reviews as reviews that self-identified as a “systematic review” or reviews that reported the sources of research and accounted for the identified studies.

Articles with inadequate outcomes; with a target population outside the age of interest (under 60 years); in a language outside the established ones—English or Portuguese; articles with other types of studies than randomized controlled trials; or systematic reviews were excluded. Studies involving food, roots, pesticide use, traditional medicines, and viral diseases were also not considered.

### 2.3. Search Strategy

The search encompassed systematic reviews and randomized controlled trials in the field of healthy aging related to exposure to nature. A structured literature search was conducted in the databases Cochrane, SCOPUS, PubMed, Web of Science, Embase, and LILACS, including all articles published in the last ten years (from January 2012 to April 2022). The search was designed to combine different nature-based interventions with aging-related outcomes (using the Boolean operators AND and OR) according to the specificity of each database. Terms were searched in Portuguese and English. The full final search strategy is provided in the Appendix A, and it was tailored to the specificities of each database used.

### 2.4. Study Selection

Two reviewers independently screened titles and abstracts for relevance to obtain the full-text articles of publications that were deemed potentially relevant by at least one reviewer. Full-text articles were screened according to predetermined exclusion criteria, and any disagreements were reconciled through team discussion. For this step, the open access application Rayyan (reference manager) was used.

### 2.5. Data Extraction

The data extracted from the selected articles were organized in a spreadsheet, with the different types of interventions and health outcomes, the reason for being included in the review, the measured outcomes, the comparators, the estimates of effect of the interventions for the outcomes of the elderly, and the characteristics of the review, among other aspects being evaluated in the spreadsheet.

### 2.6. Coding and Critical Evaluation

The next step in the development of the map involved systematically coding and extracting data using a structured methodology, along with assessing the quality of the included systematic reviews or their impact. Studies were coded according to the relevant intervention and outcome categories.

Review studies were assessed using AMSTAR2 Checklist [27]. This is a valid, reliable, and useful tool that helps differentiate systematic reviews, focusing on methodological quality and expert consensus. The aim of this instrument is to facilitate the development of high-quality reviews. The instrument contains 16 items, 7 of which are critical, which must be classified as “yes”, “partially yes”, or “no”. Depending on the number of failures in critical items, the result is the confidence level of the study among the options critically low, low, moderate, and high [27].

In the quality analysis of the included clinical trials, a tool based on CONSORT was used, which also contains 16 items that must be evaluated, with three scoring possibilities each: 0 = no description, 1 = insufficient description, and 2 = adequate description. The maximum score of 32 represents 100% compliance with the recommendations, other values, and the relative agreement [28].

### 2.7. Evidence Map

The generation of a matrix was carried out with the support of the Tableau software (Tableau Desktop Public Edition 2023.1.2), which has free access and orders and organizes the production of available knowledge on a given topic [30]. In the cells of the matrix, the circles located at the intersections between the interventions and the results represent the identified studies. In this representation, the size of the circle represents the volume of studies. The color of the circles represents the confidence level (high, moderate, and low) according to a methodological qualification of the studies that are included in the map. Hovering the cursor over a circle displays a list of the studies that the figure represents. The links to these studies lead to the full texts (if openly available) or to the records in an appropriately indicated database. This tool allows for filtering the evidence by type, country, effect (positive, negative), and intervention and is available at: https://public.tableau.com/app/profile/kaue.alves/viz/Nature-basedinterventionsonelderlypeopleshealthandwell-beinganevidencemap/EvidenceMap?publish=yes (accessed on 1 November 2023).

## 3. Results

### 3.1. Characterization of the Platform Used to Analyze the Literature Search

Rayyan is a free online application developed by Qatar Computing Research Institute (QCRI). Its storage is facilitated through the cloud and assists authors in systematic review- and meta-analysis-type research. It was developed to facilitate the process of screening abstracts and citation titles that were previously uploaded in files imported from other bibliographic management tools in various formats. The application offers a variety of features such as the inclusion of multiple contributors, labeling and filtering of citations, classification in references by included, excluded and in doubt, blinding between reviewers, and identification of potential duplicates [31].

### 3.2. Selected Studies

After applying the search strategy, 10,379 articles were retrieved. Of this total number found, 3826 were identified in the Web of Science database, 3279 in Scopus, 1523 in Pubmed, 1229 in Embase, 511 articles in Cochrane, and 11 in Lilacs.

As a first phase of evaluation, 3581 articles were removed, being classified as duplicates. In the second phase, the authors screened a total of 6798 articles, whose titles and abstracts were assessed for eligibility. Thus, for the third phase of analysis, 40 articles were available for screening through a full reading of the articles. In total, 12 studies were eligible for inclusion (see the flow diagram depicting the process of screening, Figure 1).

### 3.3. Characteristics of the Studies

The evidence map comprises 12 studies, categorized as follows: 10 randomized controlled trials and 2 systematic reviews.

The years 2019 and 2020 exhibit the highest representation in terms of the number of studies, both accounting for three studies each (*n* = 3), followed by 2012, 2014, 2016, 2017, 2018, and 2021 (*n* = 1), with a positive growth trend over the years, as shown in Figure 2.

Regarding the distribution of the year of publication vs. the type of study and intervention, the predominant intervention was forest bathing (*n* = 5), followed by walking in natural environments (*n* = 3) and therapeutic gardens (*n* = 2), according to Figure 3.

#### 3.3.1. Population

The population of interest in the study was the elderly. According to the studies surveyed, the characteristics of these older adults varied between being older adults affected by physical diseases (Chronic Obstructive Pulmonary Disease, diabetes, hypertension, and heart failure) or mental diseases (depression and Alzheimer’s) and older adults in general, without diagnosed diseases.

Among the clinical trials, the number of participants ranged from 11 to 163 participants, while for the systematic reviews, the number ranged from 231 to 930.

#### 3.3.2. Countries of the Studies

The research compiled by the map is predominantly centralized in countries in the Northern Hemisphere, including Europe, Asia, and North America, as depicted in Figure 4. Notably, there is no representation from countries in the Southern Hemisphere, encompassing South America, Africa, and Oceania.

All included articles were published in English, 11 by researchers with different professional categories within the health field and 1 by professionals in the field of architecture and engineering.

#### 3.3.3. Study Design and Quality

According to the AMSTAR 2 assessment, the two systematic reviews included were classified with a moderate confidence level. In the CONSORT assessment for randomized controlled trials, six studies also obtained a moderate confidence level and four a high confidence level, as shown in Figure 5.

#### 3.3.4. Interventions Carried out per Study

Table 1 details the operationalization, which criteria were assessed, and which possible questionnaires were used for each included article.

#### 3.3.5. Outcomes and Effects of Study

Table 2 details the outcomes and effects observed by each included paper.

#### 3.3.6. Evidence Map

Among the nature-based interventions evaluated in older adults, “forest bathing” (*n* = 5), “hiking in natural areas” (*n* = 3), “therapeutic garden” (*n* = 2), “virtual reality” (*n* = 1), and “forest sound” (*n* = 1) were surveyed. Due to the high number of types of outcomes obtained, the authors categorized the outcomes into the following categories:Physical Aspects: Cardiovascular and Pulmonary: covered by the outcomes of blood pressure reduction; heart rate reduction; heart variation reduction; vasoconstriction reduction; decreased levels of brain natriuretic peptide (BNP); improved levels of cardiovascular bioindicators (endothelin I, renin, angiotensin; Angiotensin II, angiotensin II receptor I and II); and improved lung function.Physical Aspects: Neuro-immuno-endocrinological: covered by the outcomes of decreased inflammatory response; salivary cortisol reduction; decreased levels of granzyme B and perforin; reduction in oxidative stress; healing improvement; bioimpedance improvement; reduction in agitation and mental confusion; reduction in cognitive decline; improved sleep pattern; improvement in neuropsychiatric indicators; and improved autonomy for activities of daily life.Mental/Behavioral Aspects: covered by the outcomes of depression reduction; anxiety reduction; stress reduction; tension reduction; improvement in negative emotions; increased feeling of happiness; improved quality of life; increased feeling of pleasure; increased sense of well-being; reduced feeling of fatigue; improved empathy levels.

The crossover between nature-based interventions and outcomes is described in Figure 6, which is the final evidence map. In addition to the information that is cross-referenced with interventions and outcomes, the quality analysis of the studies was described with percentage classification according to the conformity of the CONSORT and AMSTAR2 evaluation.

## 4. Discussion

The aim of this evidence map was to identify, describe, and organize the available research on nature-based interventions in older adults. With the application of the search strategy, more than 10,000 articles were identified. However, with the refinement of the search and application of the inclusion and exclusion criteria, only 12 studies were selected, pointing us to the low amount of evidence that is available on the subject.

Among the years evaluated (2012 to 2021), it is possible to see a growing trend in the number of studies published, showing that, despite the low amount of available evidence, the theme has been gaining in focus in international contexts, especially in the Northern Hemisphere, composed mostly of developed countries, without any representation of the countries of the Southern Hemisphere so far.

The evidence map was developed by crossing nature-based interventions in older adults, namely, “forest bathing”, “hiking in natural areas”, “therapeutic garden”, “virtual reality”, and “forest sound” with outcomes categorized into “Physical Aspects: Cardiovascular and Pulmonary”, “Physical Aspects: Neuro-immuno-endocrinological”, and “Mental/Behavioral Aspects”. It is an interactive, easy-to-use, and visually attractive map that summarizes the information found, also including the quality assessment of the studies.

Regarding the outcome Physical Aspects: Cardiovascular and Pulmonary, one of the notable results points to a reduction in blood pressure in four of the eleven articles analyzed. The following were cited as benefits associated with lower blood pressure: reduced heart rate, improved lung function, reduced heart failure biomarkers, reduced cardiac variation, and reduced vasoconstriction [18,32,36,37,39,40]. In the outcome Physical Aspects: Neuro-immuno-endocrinological, five studies showed positive results in reducing inflammation and activation of the immune system through plasma analysis of decreased pro-inflammatory cytokines, IL-6, IL-8, C-reactive protein, ET-1, granzyme B, tumor necrosis factor ⲁ, T lymphocytes, and NK cells [18,36,39,40,42]. The result of one study showed a reduction in biological markers of heart failure as a consequence of the attenuation of the inflammatory response and activation of antioxidant actions [40], which was also indicated in the study showing an improvement in the condition of Chronic Obstructive Disease with a gain in lung function due to an inflammatory reduction and immunological action [39]. In another study, a salivary cortisol measurement was used to assess the stress level of the elderly, and there was a reduction in the amount of cortisol after the intervention [37]. A study conducted in the Republic of Korea showed that the breathing and walking program in forest bathing brought health benefits to the elderly population in terms of neurophysiology, heart rate variability, and bioimpedance [42].

As evaluative tools used in the studies to analyze Outcomes—Mental/Behavioral Aspects, the Profile of Mood States (POMS) questionnaire, the Geriatric Depression Scale (GDS-30), Anxiety and Stress Scale (HADS), and the Perceived Stress Questionnaire (PSS) were found. The POMS was used in four studies, where reductions in the negative mood states of older adults were found [18,36,39,41].

To analyze the neuro-psychophysiological aspects, the Mini Mental State Examination, Barthel Index, and Neuropsychiatric Inventory Scale were used in one article, and improvement was found in the degree of anxiety, sleep, and quality of life in the elderly. Two articles reported an improvement in cognitive decline after an intervention in nature [34,37].

Other observed results point to reduced tension, anxiety, and fatigue [36]; increased well-being [34]; increased happiness and empathy [41]; improved negative mood states [32,40]; reduced stress [35,39]; improved sleep and quality of life [34]; and reduced depression [33,36] after the interventions.

A high heterogeneity among the articles was found on the determination of the duration of the nature-based interventions, periodicity and number of sessions, condition of the population of older people involved in the studies (healthy or unhealthy), type of nature-based intervention—with direct and indirect contact—and association of the intervention with physical activity, meditation, or breathing practices.

Regarding the duration of the interventions, they ranged from 40 min to 2 h daily (also depending on the number of weekly sessions). The literature lacks consensus regarding the ideal exposure time to nature. Two pertinent studies in the field present divergent perspectives. Researchers from the University of Exeter, United Kingdom, demonstrated that 120 min of contact with nature per week benefit physical health and psychological well-being. This study, involving nearly 20,000 individuals, recommends that two hours of exposure to natural environments, either in a single visit or through multiple brief ones, prove beneficial for diverse ethnic groups, including individuals with chronic diseases or other disabilities. The study also highlighted that proximity to natural environments near residences constituted the primary means of contact [43]. Conversely, another study from the United Kingdom suggests a shift in the paradigm from “how much time” to “how meaningful the experiences” in nature are. Therefore, it emphasizes the importance of quality moments and the appreciation of emotional interactions with the natural environment to promote well-being, focusing not on setting specific timeframes or frequencies for these moments. In essence, the emphasis lies on the meaningful connection with nature beyond mere contact [44]. Additional studies controlling for connection variables, environmental factors in the intervention settings, and even the type of intervention and its outcomes need monitoring for more conclusive insights into the topic.

Regarding the periodicity of the sessions, when applicable, variations were found between one day and seven days of weekly intervention. Interventions were also found that included a stipulation of five times a week for six months or a one-off intervention at first and a second experiment four weeks later. No linearity was found among the articles regarding periodicity, and it was not possible to establish guidance on an “ideal periodicity”.

Concerning the condition of the elderly population involved in the studies, most of them (*n* = 8) were carried out with elderly people with pre-existing diseases and conditions, such as physical diseases such as hypertension [18,36], diabetes [38], Chronic Obstructive Pulmonary Disease [39], and chronic heart failure [40,41] and mental illnesses such as depression [33] and Alzheimer’s [37].

In analogy with studies that are external to those included in the present study, there is a significant gap in the scientific literature on health promotion and disease prevention in the elderly. The specific approach aimed at this population in relation to contact with nature is scarce, and the existing studies focus mainly on pre-existing diseases and the benefits of contact with nature as a complementary treatment for these diseases [8,45].

Regarding the type of nature-based intervention performed among the studies analyzed, it is possible to observe that among the five types surveyed—forest bathing, forest sound, hiking in natural areas, therapeutic garden, and virtual reality—there is a heterogeneity of form of contact with nature. Nature-based interventions were carried out as direct contact, exemplified by forest bathing, hiking in natural areas, and therapeutic gardens, and also as indirect contact, such as forest sounds and virtual reality.

It is important to highlight that there are differences between direct contact with nature, such as forest bathing, and interventions involving indirect contact, like virtual reality. Both have their benefits and limitations and act through different mechanisms in promoting well-being. In direct contact, the sensory experience is enriched, as all senses and proprioception can be stimulated and integrated. Additionally, the role of certain volatile organic compounds and the natural microbiota of the environment seems to be associated with immune function [46].

On the other hand, there are circumstances where being physically present in nature is not possible, or for some individuals, it may not be preferable due to a dislike of insects, fear of animals present in the experience, or a greater affinity for the digital world over the natural. In such cases, alternative measures can be implemented. Virtual reality serves as a feasible intervention to provoke emotions [47]. It is expected not to have as direct an effect on the immune system, since it will not be activated in the same way. However, the psychoneuroimmunological system might have some potential for activation, although this has not been conclusively proven in research regarding this type of intervention. Yet, undoubtedly, the brain and aesthetic processing of this experience can also produce well-being [48], albeit in a more limited way, as neurosensory stimulation is restricted, since it is an artificially created natural environment mediated by technology. Nevertheless, both interventions can yield effects stemming from aesthetic appreciation, which also mediates neurotransmitters that are capable of promoting well-being [49].

As a society, there is a prejudgment that only direct and physical contact with nature brings benefits to our physical and mental health. However, research—whether or not evaluated in this study—has shown that even indirect contact, through images of nature, for example, has numerous benefits for chemotherapy patients [50], thus promoting equity of access to nature and its benefits, without necessarily being in it, for reasons of fragility due to illnesses and injuries, as in the case of many elderly people.

Finally, the survey conducted by the authors provided evidence that nature-based intervention promotes beneficial effects on the health of older adults. As a limitation of the study, the wide variation in interventions, control groups, and outcome measures among the included studies, such as duration, periodicity, and study population, limited the use of pooling of results, and this diversity contributes to a downgrading of the evidence. Nevertheless, this study offers relevant insights in several areas. In the field of management, it serves as a tool to support and develop programs and public policies that promote and encourage healthy aging, in line with the Decade of Healthy Aging, an action of the World Health Organization (2020–2030) that aims to promote a positive and active approach to aging [51]. Also, it opens space for discussions with health professionals about nature-based practices and care services, both directly and indirectly, in individual or group approaches, aiming at a holistic and equitable approach for older patients. In addition, in the research area, this study fosters the need for studies (mainly randomized controlled trials) within the theme addressed in the countries of the Southern Hemisphere, especially in Brazil, where the present study was carried out.

However, we observe that the number of randomized controlled trials on nature-based interventions is still quite limited, not only in the aging field but in other clinical contexts as well, where observational studies constitute the majority of research on this subject [52].

There are numerous challenges when conducting studies with this methodological design. While they produce a higher level of evidence and allow for discussions on cause-and-effect relationships and intervention efficacy, they are more difficult to conduct. They rely on participants’ availability to be in the field, demanding time, financial investment, and more complex logistics for operationalization.

Society’s awareness also contributes to the evolution of clinical research, yet it remains one of the greatest challenges. This sector is still underexplored and underpublicized, even among healthcare professionals. Additionally, there is widespread consensus regarding the lack of awareness concerning the effectiveness and cost of nature-based solutions (which can include nature-based health interventions) among the general public and key stakeholders, including service users and healthcare professionals. Issues of sporadic and unsustainable funding, leading to a dependence on continuity and volunteer capacity, as well as challenges in offering and maintaining urban green spaces and access to protected areas, are highlighted [53].

Natural experiments have been suggested as a robust alternative to clinical trials, especially when randomization of experiences of the intervention is unfeasible or unethical [54]. Natural experiments also offer a rigorous design and perhaps the only realistic way to measure the co-benefits of nature-based interventions in communities. However, as these natural experiments occur in real-world conditions, they are inherently complex and involve multiple layers of perspectives, values, and trade-offs. Researchers have proposed recommendations for conducting this type of investigation, emphasizing the genuine synergy between conservation biology, public health, and social science research to benefit human and planetary health. Despite the challenges in scientific research in this area, studies are necessary to advance our understanding of the efficacy of nature-based interventions, allowing for their safe implementation in public health [55].

Finally, Brazil, a country known for its immense wealth of biodiversity, becomes a favorable and rich environment for conducting research related to the theme of Nature and Health. This wealth of biodiversity enables a wide range of studies on the interactions between nature and human health. Research on the theme can cover the exposure of nature-based interventions for mental and physical well-being, in addition to indirectly impacting environmental conservations, since people who have greater proximity, familiarity, and connection with nature exhibit greater engagement in environmental conservation actions, thus contributing not only to scientific knowledge, but also to the promotion of a sustainable use of natural resources and to improving the quality of life of the elderly population.

## 5. Conclusions

Our findings show that nature-based interventions can contribute significantly to maintaining and enhancing functional abilities, ultimately leading to improved health, quality of life, and well-being in the elderly population. The insights gained from our evidence map can inform program and policy development to foster healthy aging and encourage equitable care, both in individual and group settings. It is imperative for healthcare professionals to consider the integration of nature-based practices as a valuable component of comprehensive care for older adults. Nevertheless, future research should include robustly designed randomized controlled trials to increase the evidence base for these interventions.

## Figures and Tables

**Figure 1 ijerph-21-00112-f001:**
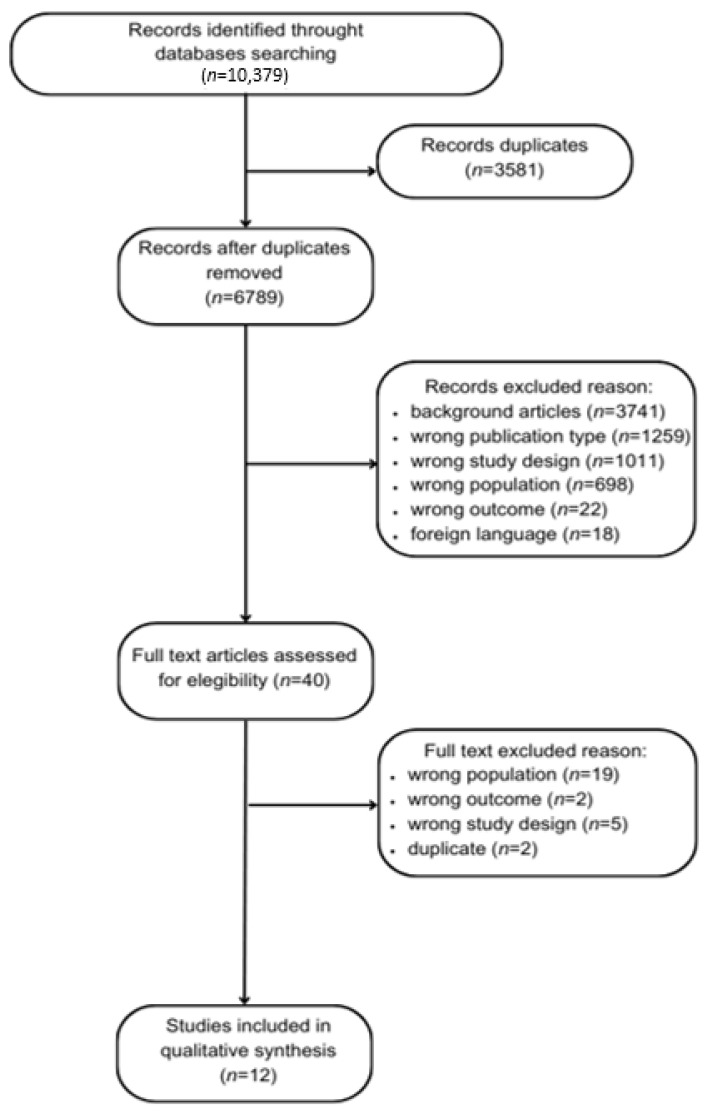
Flow diagram depicting the process of screening.

**Figure 2 ijerph-21-00112-f002:**
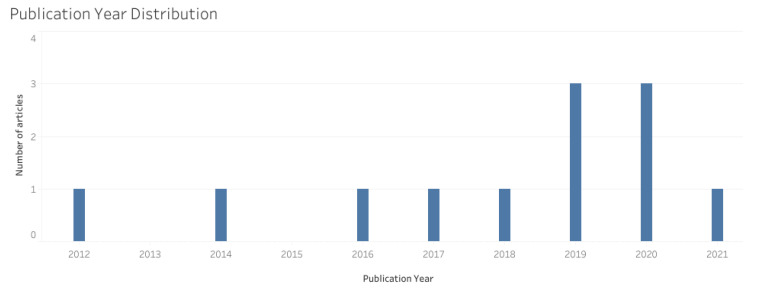
Number of articles published per year.

**Figure 3 ijerph-21-00112-f003:**
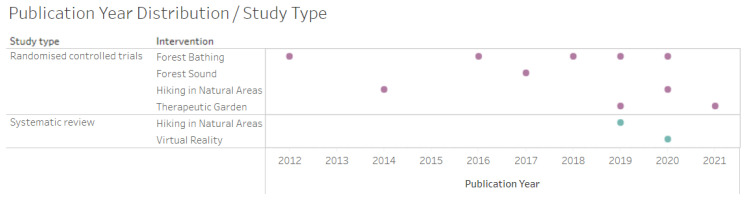
Distribution by year of publication vs. type of study and intervention (randomised controlled trials in purple and systematic review in blue).

**Figure 4 ijerph-21-00112-f004:**
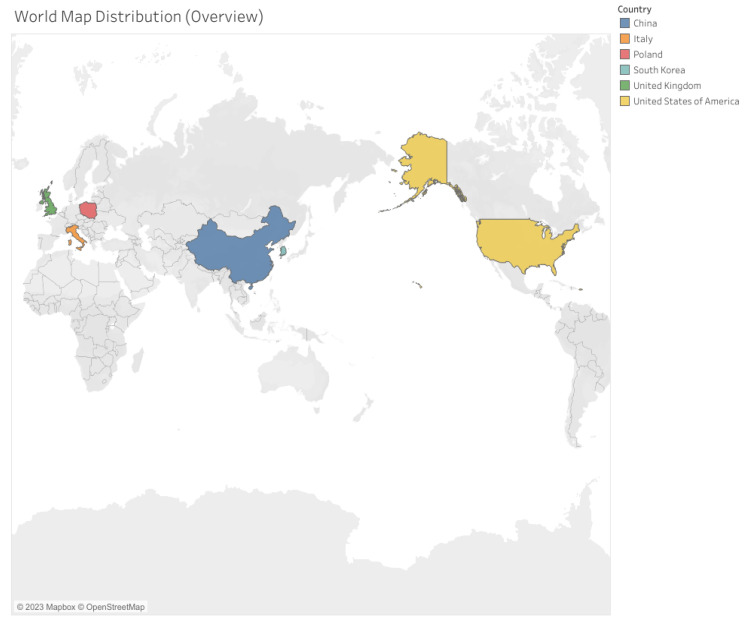
Geographic distribution of the research compiled.

**Figure 5 ijerph-21-00112-f005:**
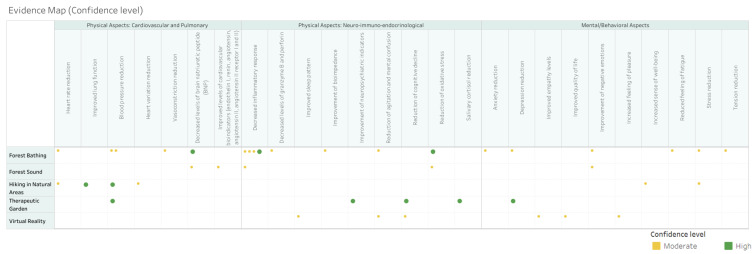
Quality rating of systematic reviews assessed by AMSTAR 2 and randomized controlled trials by CONSORT—The size of the dot corresponds to the quality (the larger, the better the article’s quality).

**Figure 6 ijerph-21-00112-f006:**
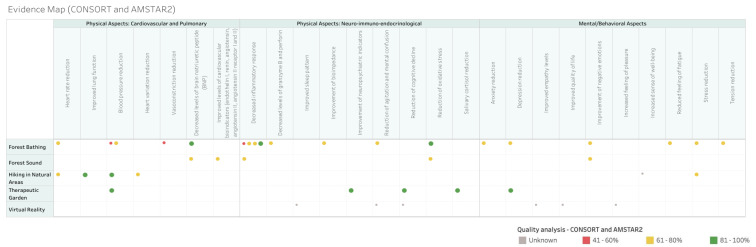
Evidence map—The size of the dot corresponds to the quality (the larger, the better the article’s quality).

**Table 1 ijerph-21-00112-t001:** Interventions by study.

N°	Author/Year	Study Type/ Sample Size	Interventions
1	Lee & Lee, 2014 [32]	RCT 70 participants	Both groups walked 1 h in the park or city and after 30 min had blood drawn. Assessed: arterial stiffness, pulmonary function, blood pressure, plus lifestyle questionnaire including smoking, alcohol consumption, and exercise. Collection pre- and post-intervention.
2	Mao et al., 2012 [18]	RCT 24 participants	Randomized into urban vs. forest. For 7 days, subjects walked a predetermined route at a calm pace for about 1.5 h, with 20 min rest. After lunch, they walked another predetermined route as well. Blood pressure, pathological factors related to cardiovascular diseases, inflammatory cytokines interleukin-6, tumor necrosis factor α, and Profile of Mood States (POMS) were assessed.
3	Szczepańska-Gieracha et al., 2021 [33]	RCT 25 participants	Control received the standard treatment (40 min general physical training and 20 min health promotion education and psychoeducation two times per week). Virtual reality (VR) group received the same treatment + VR therapy. The therapy cycle consisted of eight VR sessions of 20 min, 2× a week, for four weeks. Used Geriatric Depression Scale (GDS-30), Perception of Stress Questionnaire (PSQ), and Anxiety and Depression Scale (HADS).
4	Yeo et al., 2020 [34]	Systematic review 930 participants	Interventions of an “active” nature (intentional, direct, tactile interaction with real forms of nature or VR) vs. interventions of a “passive” nature (observation of forms of real nature, such as indoor plants) or simulated nature (nature videos). Assessment by self-reported scales, researcher observations, participant tests and tasks (e.g., to assess cognition), and direct objective measures (physiological outcomes such as pulse rate).
5	Roe et al., 2020 [35]	RCT 11 participants	Participants were randomly allocated to one of two groups, each of five to six participants. Group 1 walked the “gray” urban route on Day 1, followed by the “green” urban route on Day 2, and Group 2 vice versa, with a one-day break period between walks. Assessed Mood Adjective Check List; subjective well-being; cognitive function—reaction time; cognitive function—memory retrieval; physiological measures; real-time stress, captured with smart watch.
6	Wu et al., 2020 [36]	RCT 31 participants	The intervention group was exposed to forest bathing (*C. camphora*) vs. control in urban sites. Assessed C-reactive protein, at day 1 and 3, blood pressure measurements, O2 saturation, and heart rate before and after intervention, every day, in addition to mood state assessment.
7	Pedrinolla et al., 2019 [37]	RCT 163 participants	All patients participated in the intervention, lasting 2 h each, 5× per week for six months (120 sessions, 240 h of exposure), either in an indoor therapeutic garden (intervention group) or standard care environment (control group). Assessment by Neuropsychiatric Inventory Scale; Mini Mental State Examination; Actiheart device; Barthel Index; and Salivary cortisol.
8	Fraser et al., 2020 [38]	Systematic review 231 participants	Any form of physical activity performed in an outdoor exercise setting. Psychological assessment for depression, anxiety, quality of life, stress, general well-being.
9	Jia et al., 2016 [39]	RCT 18 participants	One group was sent to the forest (forest bath) vs. urban area (control), with no other details about the intervention. Assessed lung chemokine; surfactant protein D; interleukin-6, -8, and -1β; interferon-γ; tumor necrosis factor α; C-reactive protein; and proportion of T, NK, and POMS lymphocyte subsets.
10	Mao et al., 2018 [40]	RCT 20 participants	Randomized into urban vs. forest. After four weeks, the patients who had experienced the first forest bathing trip were recruited again, and 20 of them were enrolled for the second experiment. These 20 CHF patients were randomly categorized into two groups consisting of 10 patients in each. Collected pre- and post each experiment, fasting. Assessed: brain natriuretic peptide, interleukin-6, and tumor necrosis factor α.
11	Mao et al., 2017 [41]	RCT 33 participants	Preintervention: fasting blood draw + physical examination. Preintervention collection + POMS questionnaire. Allocated into urban vs. forest group. Subjects walked outdoors 2× per day during the experimental period, and each time, they walked along a predetermined flat path in each area at an unhurried pace for about 1.5 h. They were then asked to complete the POMS test for a second time.
12	Yi et al., 2019 [42]	RCT 88 participants	1°: Walking Program (WP) (active walking in the forest). 2°: Breathing Program (BP) (guided breathing meditation). 3°: Control group (no intervention or activities in the forest). The first two groups were conducted in urban forests. The WP consisted of 30 min of preparatory activities, 50 min of walking in the forest, 20 min of muscle training with elastic band, and 20 min of closing activities. Participants taped red Yongquan beans on both feet so that they could be stimulated by acupressure during the walk. The AP consisted of 30 min preparatory session, 30 min guided breathing meditation, 20 min slow forest walk, 20 min muscle training with elastic band, and 20 min closing activities.

RCT—randomised controlled trial.

**Table 2 ijerph-21-00112-t002:** Outcomes and effects by study.

N°	Author/Year	Outcomes and Effects
1	Lee & Lee, 2014 [32]	→ Decreased blood pressure in the forest group → Improved lung function in the forest group → The urban group did not undergo significant changes.
2	Mao et al., 2012 [18]	→ After seven days, there was a reduction in blood pressure. → After seven days, there was a reduction in IL-6 and ET-1. → Reduction of the negative subscales of the POMS questionnaire.
3	Szczepańska-Gieracha et al., 2021 [33]	→ The VR group, after eight sessions, showed a reduction in their score of the Geriatric Depression Scale-GDS 30. → Reduction in stress and anxiety in the VR group.
4	Yeo et al., 2020 [34]	→ Improvement in cognitive decline and agitation in the intervention group. → Increase in the level of empathy and feeling of happiness in the volunteers of the intervention group. → Improved performance of activities of daily living, sleep, and quality of life.
5	Roe et al., 2020 [35]	→ Lower heart rate variability in the gray urban group, which indicates higher cardiac activation and higher stress. → Accessible urban green areas offer promising opportunities for the health of older people.
6	Wu et al., 2020 [36]	→ Reduced anxiety, depression, tension, fatigue, and confusion in the forest bathing group. → Reduction in the level of diastolic blood pressure. → Reduction in C-reactive protein level in the forest group.
7	Pedrinolla et al., 2019 [37]	→ Significant improvement in the intervention group’s score on the Neuropsychiatric Inventory Scale. → Reduction in quetiapine dosage for the intervention group after the activity. → Mini Mental State Examination score was significantly better in the intervention group. → Diastolic blood pressure was lower in the intervention group. → Reduction in salivary cortisol level compared to the control group.
8	Fraser et al., 2020 [38]	→ Mixed results were found regarding quality of life, no significant effect was found for depression. → Due to the limited number of studies eligible for inclusion and the heterogeneity of outcome measures, it was difficult to draw firm conclusions.
9	Jia et al., 2016 [39]	→ In the forest group, they found significantly decreased perforin and granzyme B expressions, with decreased levels of pro-inflammatory cytokines and stress hormones. → Negative subscale scores of the Profile of Mood State questionnaire—POMS decreased after the forest bathing walk. → In patients with chronic obstructive pulmonary disease, reduced inflammation and stress level.
10	Mao et al., 2018 [40]	→ After the second 4-day forest bathing trip, decline in heart failure biomarkers and attenuation of inflammatory response and oxidative stress was observed.
11	Mao et al., 2017 [41]	→ Forest group subjects showed a significant reduction in brain natriuretic peptide. → Cardiovascular disease-related values in individuals exposed to the forest environment compared to the urban control group. → Reduced level of inflammatory cytokines and improved antioxidant function were observed in the forest group. → POMS indicated that negative emotional mood state was alleviated after intervention.
12	Yi et al., 2019 [42]	→ Both Breathing Program and Walking Program brought health benefits to the elderly population in terms of neurophysiology, heart rate variability, and bioimpedance. Beneficial effects varied depending on the characteristics of the FTPs and the types of CS. → In the study, BP was effective in increasing the phase angle of the upper limbs, and WP was effective in increasing the phase angle of the lower limbs. → WP elevated parasympathetic nervous system activity in TE-type participants. → BP was beneficial in increasing alpha or beta wave powers in type SE, and WP was effective in increasing beta wave power for type SY.

## Data Availability

The evidence map was created using Tableau software and can be fully accessed through the website: https://public.tableau.com/app/profile/kaue.alves/viz/Nature-basedinterventionsonelderlypeopleshealthandwell-beinganevidencemap/EvidenceMap?publish=yes (accessed on 1 November 2023). Other datasets generated during and/or analyzed during the current study are available upon request from the corresponding author.

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
