# Peer review of "Nature-Based Interventions Targeting Elderly People’s Health and Well-Being: An Evidence Map"

_ijerph, 2024, doi:10.3390/ijerph21010112_

Round 1
Reviewer 1 Report
Comments and Suggestions for Authors
The paper presents an interesting review on the nature-based interventions on elderly well-being.
The research is well structured and the methods and tools used are adequate.
The results are very interesting and represent a good base for future researches. As highlighted by the authors, the study has some limitations, but provides a promising basis for future investigations.
My overall recommendation is to accept the paper in present form.
Anyway, I ask the authors to make two very small interventions in their paper:
1) Line 64, authors should take a look at and integrate in the text the "Determinants of health" indicated by the WHO (see my comment in the text);
2) Line 91, authors should better clarify what do they intend with “lack scientific evidence on the topic”.

Author Response
Thank you for your thoughtful review and positive feedback on our paper. We appreciate the time and effort you dedicated to assessing our work and we are pleased to hear that you found our research on nature-based interventions for elderly well-being interesting and well-structured.
Please see the attachment.

Reviewer 2 Report
Comments and Suggestions for Authors
Overall, this is a reasonable paper that doesn’t add greatly to understanding but at least maps what has been done in terms of clinical trials in the area and (rightly) points to the need for more research, especially in the southern hemisphere, on the topic.
While it is perhaps appropriate to limit any conclusions to summarising what the studies did find, it would have been more interesting if the commentary had also been informed by a better grasp of potential theories, mechanisms and pathways by which health-relevant biomarkers indicate that nature improves health. I don’t think it appropriate to ask the authors to cover the extensive literature on these in any kind of detail, but the implications of some of this literature could perhaps be alluded to in acknowledging the limitations of the paper and in the concluding section.
In particular, I think further commentary on the use of VR in trials might be warranted given that this lacks any truly multi-sensory, embodied and biological experience of nature. Since the theoretical basis for much of the forest bathing literature posits that phytoncides and other aspects of the microbiome are implicated in the beneficial effects, it would be valuable to know if only certain kinds of benefits are found for VR-based interventions.
A brief commentary on the challenges and limitations of RCTs in the context of nature-based interventions might also be warranted, given the difficulty in undertaking research that has external ecological validity in relation to many such interventions (e.g. how public park improvements benefitting older adults living in their own homes and making their own choices about when to visit such places)
Some detailed comments:
Lines 44-45 - Portugal has “a projected 44 population decline of 22% (10.5 million to 8.6 million inhabitants) between 2012 and 2060”, yet the significance of this is not spelt out for the population and its ageing profile. Will this decline be due to older people dying (in which case the percentage of older people may stay the same) or to outward migration of working-age people (which would lead to an increased proportion of older people)?
Lines 51-53 – make clear if you are talking about the Brazilian population here or the global population.
Line 59 – make it clear that Federal Law No. 10,741 is a Brazilian Federal law.
Figure 1 – what is the difference between papers excluded for covering “wrong population type” and “wrong population”?
Line 223 says 13 studies were found eligible but Figure 1 and line 229 say 12 were included in your study. What happened to the other one?
Table 1 – it would be very helpful for the reader if the systematic review papers were clearly labelled as such and distinguished from the RCTs. It would also be helpful to have the sample size indicated for each paper.
Lines 378-381 – “Regarding the duration of the interventions, they ranged from 40 minutes to two hours daily (also depending on the number of weekly sessions). This result corroborates relevant research in the area. Conducted at the University of Exeter, one study showed that 120 minutes of contact with nature per week benefits physical health and psychological well-being.” I don’t see how the fact that the studied interventions varied in length, both of each intervention and of frequency of repetition (periodicity), corroborates the U of Exeter finding. Either more detail is needed here to show that only interventions that added up to a nature exposure of 120 minutes per week, or (if true) that some studies produced significant beneficial effects with a lower total nature exposure per week, then this needs further commentary.
Comments on the Quality of English LanguageThe English language is pretty good but needs final checking for consistency and correctness. E.g., in the abstract, “interventions on elderly’s’ health” has too many apostrophes and needs a definite article. Similarly, “In management, informs program and policy development” is not quite grammatical in English. The introductory sections (Sections 1 and 2 in particular) would benefit from editing by a native English speaker to make the points clearly and sometimes more succinctly. But the whole document needs a final check for language.
Author Response

(The authors gave the same response as above.)

Reviewer 3 Report
Comments and Suggestions for Authors
This is a well-conceived and executed study which will be very helpful to the field as we consider the needs of the aging population. The authors are to be congratulated on the quality of their work. I have minor suggestions that may improve the readability of the manuscript.
Line # Comment
29 "elderly's' " should be "elderlies' "
58 "are" should be "is"
75-76 Remove "only" from #75, insert "alone" after "areas" on #76
218 ff What is MDPI's standard punctuation for numbers. Is a hard stop "." or a comma "," used to numbers when there are more than five digits? https://community.cochrane.org/style-manual/numbers-statistics-and-units/numbers
223 Says 13 papers were included, but there were only 12.
Figure 2 and following. All figures need to be enlarged, use larger font size for the axis labels and legends. These are important figures and they were very hard to read.
Table 2 (reference #5) and elsewhere (e.g., lines 300, 345), the authors say "Reduced heart rate and heart rate variability which correlates with lower stress level." Please, clarify as low heart rate variability (HRV) is usually associated with increased stress (greater sympathetic nervous system activation) and higher HRV with increased relaxation (greater parasympathetic activation). For a quick review see: https://www.medicinenet.com/what_is_heart_rate_variability_hrv/article.htm
330 Delete "over the years", it is redundant
352 Replace "where it was shown an" with "showing"
Comments on the Quality of English Language
The quality of the English language is excellent. I have indicated minor changes in Comments and Suggestions for Authors for minor changes.
Author Response
Thank you for your thoughtful review and positive feedback on our study. We appreciate your time and attention to detail. We have carefully considered your suggestions and made the necessary revisions to enhance the clarity and accuracy of the manuscript.
Please see the attachment.

Round 2
Reviewer 2 Report
Comments and Suggestions for Authors
The authors have made appropriate amendments and improved the paper considerably by giving more attention to potential mechanisms and pathways and by acknowledging the challenges and limitations of real-world studies of nature-based interventions for older people. Well done.
I just have a few, very minor edits to suggest that I believe need attention, as follows.
In Tables 1 and 2 it would be more correct to list the papers’ authorship as, e.g. ‘Lee & Lee, 2014’ or ‘Mao et al., 2012’, rather than by first author alone, where the paper is co-authored, which I suspect is true of all the papers included in these tables.
Lines 438-450 – this discussion of VR (and its limitations in activating beneficial psychophysical responses) is welcome but some of the assertions, e.g. “undoubtedly, the brain and aesthetic processing of this experience can also produce well-being” and “aesthetic appreciation, which also mediates neurotransmitters capable of promoting well-being” need citations to supporting evidence. It’s not clear if the subsequent paragraph and reference 47 (Catissi et al., 2023) covers these points adequately but I suspect not.
Line 492 – should “Natural experiences” at the start of this second sentence in the paragraph be “Natural experiments”?
My only general comment is that I have always understood RCTs to be Randomised, Controlled Trials, not ‘Randomised Clinical Trials’ as they are called in this paper. I think this should be corrected throughout.
Author Response
Thank you for your thoughtful review and positive feedback on our study. We appreciate your time and attention to detail in this new revision. We have carefully considered your suggestions and made the necessary revisions to enhance the clarity and accuracy of the manuscript.
Please see the attachment.
